# BatchEnsemble: An alternative approach to Efficient Ensemble and Lifelong Learning

**Yeming Wen**[1,2,3*]**, Dustin Tran**[3] **& Jimmy Ba**[1,2]
[1]University of Toronto, [2]Vector Institute, [3]Google Brain

## Abstract

Ensembles, where multiple neural networks are trained individually and their predictions are averaged, have been shown to be widely successful for improving both the accuracy and predictive uncertainty of single neural networks. However, an ensemble's cost for both training and testing increases linearly with the number of networks, which quickly becomes untenable.

In this paper, we propose BatchEnsemble[1], an ensemble method whose computational and memory costs are significantly lower than typical ensembles. BatchEnsemble achieves this by defining each weight matrix to be the Hadamard product of a shared weight among all ensemble members and a rank-one matrix per member. Unlike ensembles, BatchEnsemble is not only parallelizable across devices, where one device trains one member, but also parallelizable within a device, where multiple ensemble members are updated simultaneously for a given mini-batch. Across CIFAR-10, CIFAR-100, WMT14 EN-DE/EN-FR translation, and out-of-distribution tasks, BatchEnsemble yields competitive accuracy and uncertainties as typical ensembles; the speedup at test time is 3X and memory reduction is 3X at an ensemble of size 4. We also apply BatchEnsemble to lifelong learning, where on Split-CIFAR-100, BatchEnsemble yields comparable performance to progressive neural networks while having a much lower computational and memory costs. We further show that BatchEnsemble can easily scale up to lifelong learning on Split-ImageNet which involves 100 sequential learning tasks.

## 1 Introduction

Ensembling is one of the oldest tricks in machine learning literature (Hansen & Salamon, 1990). By combining the outputs of several models, an ensemble can achieve better performance than any of its members. Many researchers demonstrate that a good ensemble is one where the ensemble's members are both accurate and make independent errors (Perrone & Cooper, 1992; Maclin & Opitz, 1999). In neural networks, SGD (Bottou, 2003) and its variants such as Adam (Kingma & Ba, 2014) are the most common optimization algorithm. The random noise from sampling mini-batches of data in SGD-like algorithms and random initialization of the deep neural networks, combined with the fact that there is a wide variety of local minima solutions in high dimensional optimization problem (Ge et al., 2015; Kawaguchi, 2016; Wen et al., 2019), results in the following observation: deep neural networks trained with different random seeds can converge to very different local minima although they share similar error rates. One of the consequence is that neural networks trained with different random seeds will usually not make all the same errors on the test set, i.e. they may disagree on a prediction given the same input even if the model has converged (Fort et al., 2019).

Ensembles of neural networks benefit from the above observation to achieve better performance by averaging or majority voting on the output of each ensemble member (Xie et al., 2013; Huang et al., 2017). It is shown that ensembles of models perform at least as well as its individual members and diverse ensemble members lead to better performance (Krogh & Vedelsby, 1995). More recently, Lakshminarayanan et al. (2017) showed that deep ensembles give reliable predictive uncertainty estimates while remaining simple and scalable. A further study confirms that deep ensembles generally

---

*Partial work done as part of the Google Student Researcher Program. Email: ywen@cs.toronto.edu

[1]https://github.com/google/edward2

achieves the best performance on out-of-distribution uncertainty benchmarks (Ovadia et al., 2019; Gustafsson et al., 2019) compared to other methods such as MC-dropout (Gal & Ghahramani, 2015).

Despite their success on benchmarks, ensembles are limited in practice due to their expensive computational and memory costs, which increase linearly with the ensemble size in both training and testing. Computation-wise, each ensemble member requires a separate neural network forward pass of its inputs. Memory-wise, each ensemble member requires an independent copy of neural network weights, each up to millions (sometimes billions) of parameters. This memory requirement also makes many tasks beyond supervised learning prohibitive. For example, in lifelong learning, a natural idea is to use a separate ensemble member for each task, adaptively growing the total number of parameters by creating a new independent set of weights for each new task. No previous work achieves competitive performance on lifelong learning via ensemble methods, as memory is a major bottleneck.

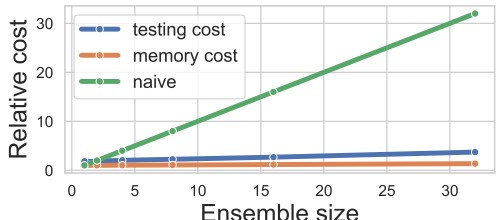

Figure 1: The test time cost (blue) and memory cost of BatchEnsemble (orange) w.r.t the ensemble size. The result is relative to single model cost. Testing time cost and memory cost of naive ensemble are plotted in green.

**Our contribution**: In this paper, we aim to address the computational and memory bottleneck by building a more parameter efficient-ensemble method: BatchEnsemble. We achieve this goal by exploiting a novel ensemble weight generation mechanism: the weight of each ensemble member is generated by the Hadamard product between: **a.** one shared weight among all ensemble members. **b.** one rank-one matrix that varies among all members, which we refer to as fast weight in the following sections. Figure 1 compares testing and memory cost between BatchEnsemble and naive ensemble. Unlike typical ensembles, BatchEnsemble is mini-batch friendly, where it is not only parallelizable across devices like typical ensembles but also parallelizable within a device. Moreover, it incurs only minor memory overhead because a large number of weights are shared across ensemble members.

Empirically, we show that BatchEnsemble has the best trade-off among accuracy, running time, and memory on several deep learning architectures and learning tasks: CIFAR-10/100 classification with ResNet32 (He et al., 2016) and WMT14 EN-DE/EN-FR machine translation with Transformer (Vaswani et al., 2017). Additionally, we show that BatchEnsemble is effective in calibrated prediction on out-of-distribution datasets; and uncertainty evaluation on contextual bandits. Finally, we show that BatchEnsemble can be successfully applied in lifelong learning and scale up to 100 sequential learning tasks without catastrophic forgetting and the need of a memory buffer. Section 5 further provides diversity analysis as a tool to understand why BatchEnsemble works well in practice.

## 2 BACKGROUND

In this section, we describe relevant background about ensembles, uncertainty evaluation, and lifelong learning for our proposed method, BatchEnsemble.

### 2.1 ENSEMBLES FOR IMPROVED PERFORMANCE

Bagging, also called bootstrap aggregating, is an algorithm to improve the total generalization performance by combining several different models (Breiman, 1996). Strategies to combine those models such as averaging and majority voting are known as ensemble methods. It is shown that ensembles of models perform at least as well as each of its ensemble members (Krogh & Vedelsby, 1995). Moreover, ensembles achieve the best performance when each of their members makes independent errors (Goodfellow et al., 2015; Hansen & Salamon, 1990).

**Related work on ensembles**: Ensembles have been studied extensively for improving model performance (Hansen & Salamon, 1990; Perrone & Cooper, 1992; Dietterich, 2000; Maclin & Opitz, 1999). One major direction in ensemble research is how to reduce their cost at test time. Bucila et al. (2006) developed a method to compress large, complex ensembles into smaller and faster models which achieve faster test time prediction. Hinton et al. (2015) developed the above approach further by distilling the knowledge in an ensemble of models into one single neural network. Another

major direction in ensemble research is how to reduce their cost at training time. Xie et al. (2013) forms ensembles by combining the output of networks within a number of training checkpoints, named Horizontal Voting Vertical Voting and Horizontal Stacked Ensemble. Additionally, models trained with different regularization and augmentation can be used as ensemble to achieve better performance in semi-supervised learning (Laine & Aila, 2017). More recently, Huang et al. (2017) proposed Snapshot ensemble, in which a single model is trained by cyclic learning rates (Loshchilov & Hutter, 2016; Smith, 2015) so that it is encouraged to visit multiple local minima. Those local minima solutions are then used as ensemble members. Garipov et al. (2018) proposed fast geometric ensemble where it finds modes that can be connected by simple curves, and each mode can taken as one ensemble member. The aforementioned works are complementary to BatchEnsemble, and one could potentially combine these techniques to achieve better performance. BatchEnsemble is efficient in both computation (including training and testing) and memory, along with a minimal change to the current training scheme such as learning rate schedule. For example, the need of cyclic learning rates in Snapshot Ensemble makes it incompatible to Transformer (Vaswani et al., 2017) which requires a warm-up and inverse square root learning rate.

Explicit ensembles are expensive so another line of work lies on what so-called "implicit" ensembles. For example, Dropout (Srivastava et al., 2014) can be interpreted as creating an exponential number of weight-sharing sub-networks, which are implicitly ensembled in test time prediction (Warde-Farley et al., 2014). MC-dropout can be used for uncertainty estimates (Gal & Ghahramani, 2015). Implicit ensemble methods are generally cost-free in training and testing.

## 2.2    Ensembles for Improved Uncertainty

Several measures have been proposed to assess the quality of uncertainty estimates, such as calibration (Dawid, 1982; Degroot & Fienberg, 1983). Another important metric is the generalization of predictive uncertainty estimates to out-of-distribution datasets (Hendrycks & Dietterich, 2019). The contextual bandits task was recently proposed to evaluate the quality of predictive uncertainty, where maximizing reward is of direct interest (Riquelme et al., 2018); and which requires good uncertainty estimates in order to balance exploration and exploitation.

Although deep neural networks achieve state-of-the-art performance on a variety of tasks, their predictions are often poorly calibrated (Guo et al., 2017). Bayesian neural networks (Hinton & Neal, 1995), which posit a distribution over the weights rather than a point estimate, are often used for model uncertainty (Dusenberry et al., 2019). However, they require modifications to the traditional neural network training scheme. Deep ensembles have been proposed as a simple and scalable alternative, and have been shown to make well-calibrated uncertainty estimates (Lakshminarayanan et al., 2017). More recently, Ovadia et al. (2019) and Gustafsson et al. (2019) independently benchmarked existing methods for uncertainty modelling on a broad range of datasets and architectures, and observed that ensembles tend to outperform variational Bayesian neural networks in terms of both accuracy and uncertainty, particularly on OOD datasets. Fort et al. (2019) investigates the loss landscape and postulates that variational methods only capture local uncertainty whereas ensembles explore different global modes. It explains why deep ensembles generally perform better.

## 2.3    Lifelong Learning

In lifelong learning, the model trains on a number of tasks in a sequential (online) order, without access to entire previous tasks' data (Thrun, 1998; Zhao & Schmidhuber, 1996). One core difficulty of lifelong learning is "catastrophic forgetting": neural networks tend to forget what it has learnt after training on the subsequent tasks (McCloskey, 1989; French, 1999). Previous work on alleviating catastrophic forgetting can be divided into two categories.

In the first category, updates on the current task are regularized so that the neural network does not forget previous tasks. Elastic weight consolidation (EWC) applies a penalty on the parameter update based on the distance between the parameters for the new and the old task using the Fisher information metric (Kirkpatrick et al., 2016). Other methods maintain a memory buffer that stores a number of data points from previous tasks. For example, gradient episodic memory approach penalizes the gradient on the current task so that it does not increase the loss of examples in the memory buffer (Lopez-Paz & Ranzato, 2017; Chaudhry et al., 2018). Another approach combines experience replay algorithms with lifelong learning (Rolnick et al., 2018; Riemer et al., 2018).

In the second category, one increases model capacity as new tasks are added. For example, progressive neural networks (PNN) copy the entire network for the previous task and add new hidden units when adopting to a new task (Rusu et al., 2016). This prevents forgetting on previous tasks by construction (the network on previous tasks remains the same). However, it leads to significant memory consumption when faced with a large number of lifelong learning tasks. Some following methods expand the model in a more parameter efficient way at the cost of introducing an extra learning task and not entirely preventing forgetting. Yoon et al. (2017) applies group sparsity regularization to efficiently expand model capacity; Xu & Zhu (2018) learns to search for the best architectural changes by carefully designed reinforcement learning strategies.

## 3 METHODS

As described above, ensembles suffer from expensive memory and computational costs. In this section, we introduce BatchEnsemble, an efficient way to ensemble deep neural networks.

### 3.1 BATCHENSEMBLE

In this section, we introduce how to ensemble neural networks in an efficient way. Let $W$ be the weights in a neural network layer. Denote the input dimension as $m$ and the output dimension as $n$, i.e. $W \in \mathbb{R}^{m \times n}$. For ensemble, assuming the ensemble size is $M$ and each ensemble member has weight matrix $\overline{W}_i$. Each ensemble member owns a tuple of trainable vectors $r_i$ and $s_i$ which share the same dimension as output and input ($n$ and $m$) respectively, where $i$ ranges from 1 to $M$. Our algorithm generates a family of ensemble weights $\overline{W}_i$ by the following:

$$\overline{W}_i = W \circ F_i, \text{ where } F_i = s_i r_i^\top, \quad (1)$$

For each training example in the mini-batch, it receives an ensemble weight $\overline{W}_i$ by element-wise multiplying $W$, which we refer to as "slow weights", with a rank-one matrix $F_i$, which we refer to as "fast weights." The subscript $i$ represents the selection of ensemble member. Since $W$ is shared across ensemble members, we term it as "shared weight" in the following paper. Figure 2 visualizes BatchEnsemble. Rather than modulating the weight matrices, one can also modulate the neural networks' intermediate features, which achieves promising performance in visual reasoning tasks (Perez et al., 2017).

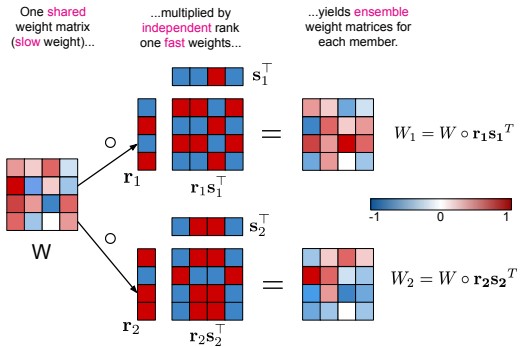

Figure 2: An illustration on how to generate the ensemble weights for two ensemble members.

**Vectorization**: We show how to make the above ensemble weight generation mechanism parallelizable within a device, i.e., where one computes a forward pass with respect to multiple ensemble members in parallel. This is achieved by manipulating the matrix computations for a mini-batch (Wen et al., 2018). Let $x$ denote the activations of the incoming neurons in a neural network layer. The next layer's activations are given by:

$$y_n = \phi\left(\overline{W}_i^\top x_n\right) \quad (2)$$

$$= \phi\left(\left(W \circ s_i r_i^\top\right)^\top x_n\right) \quad (3)$$

$$= \phi\left(\left(W^\top (x_n \circ s_i)\right) \circ r_i\right), \quad (4)$$

where $\phi$ denotes the activation function and the subscript $n$ represents the index in the mini-batch. The output represents next layer's activations from the $i^{th}$ ensemble member. To vectorize these computations, we define matrices $R$ and $S$ whose rows consist of the vectors $r_i$ and $s_i$ for all examples in the mini-batch. The above equation is vectorized as:

$$Y = \phi\left(\left((X \circ S)W\right) \circ R\right). \quad (5)$$

where $X$ is the mini-batch input. By computing Eqn. 5, we can obtain the next layer's activations for each ensemble member in a mini-batch friendly way. This allows us to take full advantage of parallel accelerators to implement the ensemble efficiently. To match the input and the ensemble weight, we can divide the input mini-batch into $M$ sub-batches and each sub-batch receives ensemble weight $\overline{W}_i, i = \{1, \ldots, M\}$.

**Ensembling During Testing**: In our experiments, we take the average of predictions of each ensemble member. Suppose the test batch size is $B$ and there are $M$ ensemble members. To achieve an efficient implementation, one repeats the input mini-batch $M$ times, which leads to an effective batch size $B \cdot M$. This enables all ensemble members to compute the output of the same $B$ input data points in a single forward pass. It eliminates the need to calculate the output of each ensemble member sequentially and therefore reduces the ensemble's computational cost.

## 3.2    Computational Cost

The only extra computation in BatchEnsemble over a single neural network is the Hadamard product, which is cheap compared to matrix multiplication. Thus, BatchEnsemble incurs almost no additional computational overhead (Figure 1).[2] One limitation of BatchEnsemble is that if we keep the mini-batch size the same as single model training, each ensemble member gets only a portion of input data. In practice, the above issue can be remedied by increasing the batch size so that each ensemble member receives the same amount of data as ordinary single model training. Since BatchEnsemble is parallelizable within a device, increasing the batch size incurs almost no computational overhead in both training and testing stages on the hardware that can fully utilize large batch size. Moreover, when increasing the batch size reaches its diminishing return regime, BatchEnsemble can still take advantage from even larger batch size by increasing the ensemble size.

The only memory overhead in BatchEnsemble is the set of vectors, $\{r_1, \ldots, r_m\}$ and $\{s_1, \ldots, s_m\}$, which are cheap to store compared to the weight matrices. By eliminating the need to store full weight matrices of each ensemble member, BatchEnsemble has almost no additional memory cost. For example, BatchEnsemble of ResNet-32 of size 4 incurs $10\%$ more parameters while naive ensemble incurs 3X more.

## 3.3    BatchEnsemble as an Approach to lifelong learning

The significant memory cost of ensemble methods limits its application to many real world learning scenarios such as multi-task learning and lifelong learning, where one might apply an independent copy of the model for each task. This is not the case with BatchEnsemble. Specifically, consider a total of $T$ tasks arriving in sequential order. Denote $D_t = (x_i, y_i, t)$ as the training data in task $t$ where $t \in \{1, 2, \ldots, T\}$ and $i$ is the index of the data point. Similarly, denote the test data set as $\mathcal{T}_t = (x_i, y_i, t)$. At test time, we compute the average performance on $\mathcal{T}_t$ across all tasks seen so far as the evaluation metric. To extend BatchEnsemble to lifelong learning, we compute the neural network prediction in task $t$ with weight $\overline{W}_t = W \circ (r_t s_t^\top)$ in task $t$. In other words, each ensemble member is in charge of one lifelong learning task. For the training protocol, we train the shared weight $W$ and two fast weights $r_1, s_1$ on the first task,

$$\min_{W, s_1, r_1} L_1(W, s_1, r_1; D_1), \tag{6}$$

where $L_1$ is the objective function in the first task such as cross-entropy in image classification. On a subsequent task $t$, we only train the relevant fast weights $r_t, s_t$.

$$\min_{s_t, r_t} L_t(s_t, r_t; D_t). \tag{7}$$

BatchEnsemble shares similar advantages as progressive neural networks (PNN): it entirely prevents catastrophic forgetting as the model for previously seen tasks remains the same. This removes the need of storing any data from previous task. In addition, BatchEnsemble has significantly less memory consumption than PNN as only fast weights are trained to adapt to a new task. Therefore, BatchEnsemble can easily scale to up to 100 tasks as we showed in Section 4.1 on split ImageNet. Another benefit of BatchEnsemble is that if future tasks arrive in parallel rather than sequential order,

---

[2]In Figure 1, note the computational overhead of BatchEnsemble at the ensemble size 1 indicates the additional cost of Hadamard products.

Table 1: Computational and memory costs on Split-CIFAR100 on LeNet. Numbers are relative to vanilla neural network.

|               | Vanilla | BatchE | DEN  | PNN  | RCL   |
|---------------|---------|--------|------|------|-------|
| Computational | 1       | **1.11** | 9.58 | 1.12 | 26.41 |
| Memory        | 1       | **1.10** | 5.31 | 4.16 | 2.52  |

one can train on all the tasks at once (see Section 3.1). We are not aware of any other lifelong learning methods can achieve this.

**Limitations**: BatchEnsemble is one step toward toward a full lifelong learning agent that is both immune to catastrophic forgetting and parameter-efficient. On existing benchmarks like split-CIFAR and split-ImageNet, Section 4.1 shows that BatchEnsemble's rank-1 perturbation per layer provides enough expressiveness for competitive state-of-the-art accuracies. However, one limitation of BatchEnsemble is that only rank-1 perturbations are fit to each lifelong learning task and thus the model's expressiveness is a valid concern when each task is significantly varied. Another limitation is that the shared weight is only trained on the first task. This implies that only information learnt for the first task can transfer to subsequent tasks. There is no explicit transfer, for example, between the second and third tasks. One solution is to enable lateral connections to features extracted by the weights of previously learned tasks, as done in PNN. However, we found that no lateral connections were needed for Split-CIFAR100 and Split-ImageNet. Therefore we leave the above solution to future work to further improve BatchEnsemble for lifelong learning.

**Computational cost compared to other methods**: Dynamically expandable networks (Yoon et al., 2017) and reinforced continual learning (Xu & Zhu, 2018) are two recently proposed lifelong learning methods that achieve competitive performance. These two methods can be seen as an improved version progressive neural network (PNN) (Rusu et al., 2016) in terms of memory efficiency. As shown in Xu & Zhu (2018), all three methods result to similar accuracy measure in Split-CIFAR100 task. Therefore, among three evaluation metrics (accuracy, forgetting and cost), we only compare the accuracy of BatchEnsemble to PNN in Section 4.1 and compare the cost in this section. We first compute the cost relative to PNN on Split-CIFAR100 on LeNet and then compute the rest of the numbers base on what were reported in Xu & Zhu (2018). Notice that PNN has no much computational overhead on Split-CIFAR100 because the number of total tasks is limited to 10. Even on the simple setup above, BatchEnsemble gives the best computational and memory efficiency. BatchEnsemble leads to more lower costs on large lifelong learning tasks such as Split-ImageNet.

## 4 EXPERIMENTS

Section 4.1 first demonstrates BatchEnsemble's effectiveness as an alternative approach to lifelong learning on Split-CIFAR and Split-ImageNet. Section 4.2 and Section 4.3 next evaluate BatchEnsemble on several benchmark datasets with common deep learning architectures, including image classification with ResNet (He et al., 2016) and neural machine translation with Transformer (Vaswani et al., 2017). Section 4.4 demonstrates that BatchEnsemble can be used for calibrated prediction. Finally, we showcase its applications in uncertainty modelling in Appendix C and Appendix D. Detailed description of datasets we used is in Appendix A. Implementation details are in Appendix B.

### 4.1 LIFELONG LEARNING

We showcase BatchEnsemble for lifelong learning on Split-CIFAR100 and Split-ImageNet. Split-CIFAR100 proposed in Rebuffi et al. (2016) is a harder lifelong learning task than MNIST permutations and MNIST rotations (Kirkpatrick et al., 2016), where one introduces a new set of classes upon the arrival of a new task. Each task consists of examples from a disjoint set of $100/T$ classes assuming $T$ tasks in total. To show that BatchEnsemble is able to scale to 100 sequential tasks, we also build our own Split-ImageNet dataset which shares the same property as Split-CIFAR100 except more classes (and thus more tasks) and higher image resolutions are involved. More details about these two lifelong learning datasets are provided in Appendix A.

We consider $T = 20$ tasks on Split-CIFAR100, following the setup of Lopez-Paz & Ranzato (2017). We used ResNet-18 with slightly fewer number of filters across all convolutional layers. Note that for

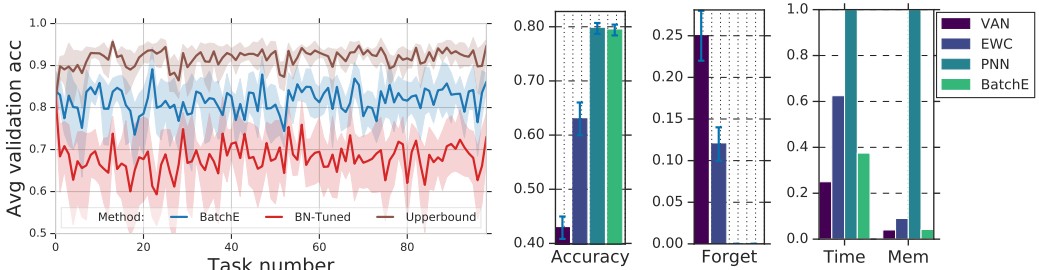

(a) Averaged validation accuracy on Split-ImageNet          (b) Various of measures on Split-CIFAR100

Figure 3: Performance for lifelong learning. **(a)**: Validation accuracy for each Split-ImageNet task. Standard deviation is computed over 5 random seeds. **(b)**: BatchEnsemble and several other methods on Split-CIFAR100. BatchEnsemble achieves the best trade-off among **Accuracy** ($\uparrow$), **Forget** ($\downarrow$), and **Time & Memory** ($\downarrow$) costs. **VAN**: Vanilla neural network. **EWC**: Elastic weight consolidation (Kirkpatrick et al., 2016). **PNN**: Progressive neural network (Rusu et al., 2016). **BN-Tuned**: Fine tuning Batch Norm layer per subsequent tasks. **BatchE**: BatchEnsemble. **Upperbound**: Individual ResNet-50 per task.

the purpose of making use of the task descriptor, we build a different final dense layer per task. We compare BatchEnsemble to progressive neural networks (PNN) (Rusu et al., 2016), vanilla neural networks, and elastic weight consolidation (EWC) on Split-CIFAR100. Xu & Zhu (2018) reported similar accuracies among DEN (Yoon et al., 2017), RCL (Xu & Zhu, 2018) and PNN. Therefore we compare accuracy only to PNN which has an official implementation and only compare computational and memory costs to DEN and RCL in Table 1.

Figure 3b displays results on Split-CIFAR100 over three metrics including accuracy, forgetting, and cost. The accuracy measures the average validation accuracy over total 20 tasks after lifelong learning ends. Average forgetting over all tasks is also presented in Figure 3b. Forgetting on task $t$ is measured by the difference between accuracy of task $t$ right after training on it and at the end of lifelong learning. It measures the degree of catastrophic forgetting. As showed in Figure 3b, BatchEnsemble achieves comparable accuracy as PNN while having 4X speed-up and 50X less memory consumption. It also preserves the no-forgetting property of PNN. Therefore BatchEnsemble has the best trade-off among all compared methods.

For Split-ImageNet, we consider $T = 100$ tasks and apply ResNet-50 followed by a final linear classifier per task. The parameter overhead of BatchEnsemble on Split-ImageNet over 100 sequential tasks is 20%: the total number of parameters is 30M v.s. 25M (vanilla ResNet-50). PNN is not capable of learning 100 sequential tasks due to the significant memory consumption; other methods noted above have also not shown results at ImageNet scale. Therefore we adopt two of our baselines. The first baseline is "BN-Tuned", which fine-tunes batch normalization parameters per task and which has previously shown strong performance for multi-task learning (Mudrakarta et al., 2018). To make a fair comparison, we augment the number of filters in BN-Tuned so that both methods have the same number of parameters. The second baseline is a naive ensemble which trains an individual ResNet-50 per task. This provides a rough upper bound on the BatchEnsemble's expressiveness per task. Note BatchEnsemble and both baselines are immune to catastrophic forgetting. So we consider validation accuracy on each subsequent task as evaluation metric. Figure 3a shows that BatchEnsemble outperforms BN-Tuned consistently. This demonstrates that BatchEnsemble is a practical method for lifelong learning that scales to a large number of sequential tasks.

## 4.2 MACHINE TRANSLATION

In this section, we evaluate BatchEnsemble on the Transformer (Vaswani et al., 2017) and the large-scale machine translation tasks WMT14 EN-DE/EN-FR. We apply BatchEnsemble to all self-attention layers with an ensemble size of 4. The ensemble in a self-attention layer can be interpreted as each ensemble member keeps their own attention mechanism and makes independent decisions. We conduct our experiments on WMT16 English-German dataset and WMT14 English-French dataset with Transformer base (65M parameters) and Transformer big (213M parameters). We maintain

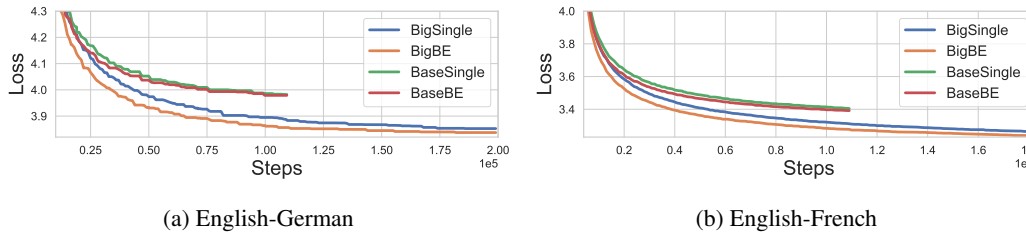

(a) English-German                               (b) English-French

Figure 4: Comparison between BatchEnsemble and single model on WMT English-German and English-French. Training stops after the model reaches targeted validation perplexity. BatchEnsemble gives a faster convergence by taking the advantage of multiple models. (a): Validation loss of WMT16 English-German task. (b): Validation loss of WMT14 English-French task. **Big**: Tranformer big model. **Base**: Transformer base model. **BE**: BatchEnsemble. **Single**: Single model.

exactly the same training scheme and hyper-parameters between single Transformer model and BatchEnsemble Transformer model.

As the result shown in Figure 4, BatchEnsemble achieves a much faster convergence than a single model. Big BatchEnsemble Transformer is roughly 1.5X faster than single big Transformer on WMT16 English-German. In addition, the BatchEnsemble Transformer also gives a lower validation perplexity than big Transformer (Table 2). This suggests that BatchEnsemble is promising for bigger Transformer models. We also compared BatchEnsemble

Table 2: Perplexity on Newstest2013 with big Transformer. BatchEnsemble with ensemble size 4.

|       | Single | MC-drop | BatchE |
|-------|--------|---------|--------|
| EN-DE | 4.30   | 4.30    | **4.26** |
| EN-FR | 2.76   | 2.77    | **2.74** |

to dropout ensemble (MC-drop in Table 2). Transformer single model itself uses dropout layers. We run multiple forward passes with different sampled dropout maskd during testing. The sample size is 16 which is already 16X more expensive than BatchEnsemble. As Table 2 showed, dropout ensemble doesn't give better performance than single model. However, Appendix B shows that while BatchEnemble's test BLEU score increases faster over the course of training, BatchEnsemble which gives lower validation loss does not achieve a better BLEU score over the single model.

## 4.3 CLASSIFICATION

We evaluate BatchEnsemble on classification tasks with CIFAR-10/100 dataset (Krizhevsky, 2009). We run our evaluation on ResNet32 (He et al., 2016). To achieve 100% training accuracy on CIFAR100, we use 4X more filters than the standard ResNet-32. In this section, we compare to MC-dropout (Gal & Ghahramani, 2015), which is also a memory efficient ensemble method. We add one more dense layer fol-

Table 3: Validation accuracy on ResNet32. Ensemble with size 4. MC-drop stands for Dropout ensemble (Gal & Ghahramani, 2015).

|      | Single | MC-drop | BatchE | NaiveE |
|------|--------|---------|--------|--------|
| C10  | 95.31  | 95.72   | **95.94** | 96.30  |
| C100 | 78.32  | 78.89   | **80.32** | 81.02  |

lowed by dropout before the final linear classifier so that the number of parameters of MC-dropout are the same as BatchEnsemble. Most hyper-parameters are shared across the single model, BatchEnsemble, and MC-dropout. More details about hyper-parameters are in Appendix B. Note that we increase the training iterations for BatchEnsemble to reach its best performance because each ensemble member gets only a portion of input data.

We train both BatchEnsemble model and MC-dropout with 375 epochs on CIFAR-10/100, which is 50% more iterations than single model. Although the training duration is longer, BatchEnsemble is still significantly faster than training individual model sequentially. Another implementation that leads to the same performance is to increase the mini-batch size. For example, if we use 4X large mini-batch size then there is no need to increase the training iterations. Table 3 shows that BatchEnsemble reaches better accuracy than single model and MC-dropout. We also calculate the accuracy of naive ensemble, whose members consist of individually trained single models. Its accuracy can be viewed as the upper bound of efficient ensembling methods. For fairness, we also compare BatchEnsemble to naive ensemble of small models in Appendix F.

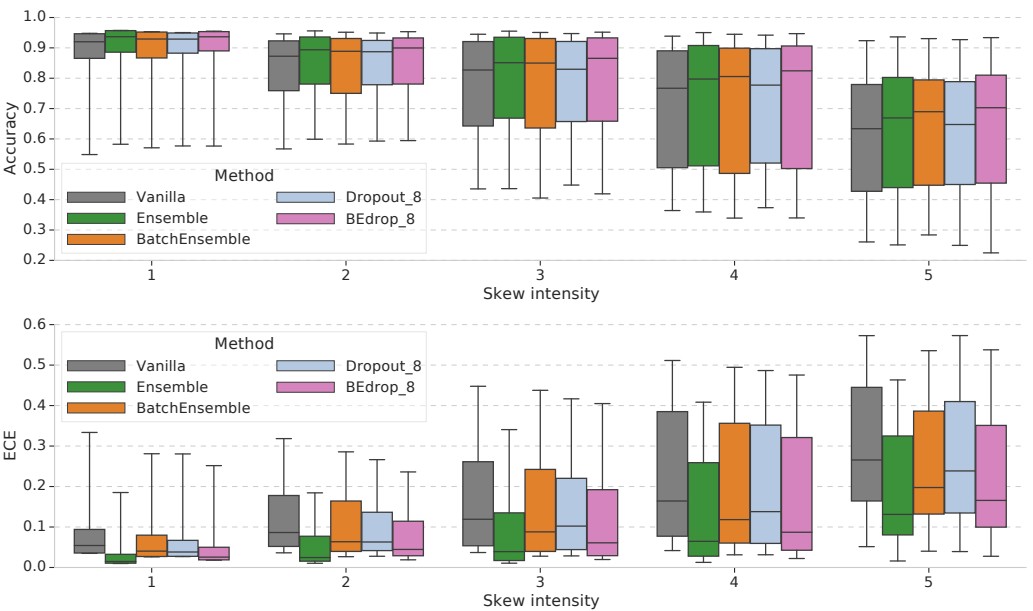

Figure 5: Calibration on CIFAR-10 corruptions: boxplots showing a comparison of ECE under all types of corruptions on CIFAR-10. Each box shows the quartiles summarizing the results across all types of skew while the error bars indicate the min and max across different skew types. **Ensemble/BatchEnsemble:** Naive/Batch ensemble of 4 ResNet32x4 models. **Dropout-8:** Dropout ensemble with sample size 8. **BEDrop-8:** BatchEnsemble of 4 models + Dropout ensemble with sample size 8. A similar measurement can be found in Ovadia et al. (2019).

## 4.4    CALIBRATION ON CORRUPTED DATASET

In this section, we measure the calibrated prediction of BatchEnsemble on corrupted datasets. Other uncertainty modelling tasks such as contextual bandits are delegated to Appendix C and Appendix D.

Other than unseen classes, corruption is another type of out-of-distribution examples. It is common that the collected data is corrupted or mislabelled. Thus, measuring uncertainty modelling under corruption is practically meaningful. We want our model to preserve uncertainty or calibration in this case. In this section, we evaluate the calibration of different methods on recently proposed CIFAR-10 corruption dataset (Hendrycks & Dietterich, 2019). The dataset consists of over 30 types of corruptions to the images. Notice that the corrupted dataset is used as a testset without training on it. Given the predictions on CIFAR-10 corruption, we can compare accuracy and calibration measure such as ECE loss for single neural network, naive ensemnble, and BatchEnsemble. Ovadia et al. (2019) benchmarked a number of methods on CIFAR-10 corruption. Their results showed that naive ensemble achieves the best performance on both accuracy and ECE loss, outperforming other methods including dropout ensemble, temperature scaling and variational methods significantly. Dropout ensemble is the state-of-the-art memory efficient ensemble method.

The scope of this paper is on efficient ensembles. Thus, in this section, we mainly compare BatchEnsemble to dropout ensemble on CIFAR-10 corruption. Naive ensemble is also plotted as an upper bound of our method. As showed in Figure 5, BatchEnsemble and dropout ensemble achieve comparable accuracy on corrupted dataset on all skew intensities. Calibration is a more important metric than accuracy when the dataset is corrupted. We observed that BatchEnsemble achieves better average calibration than dropout as the skew intensity increases. Moreover, dropout ensemble requires multiple forward passes to get the best performance. Ovadia et al. (2019) used sample size 128 while we found no significant difference between sample size 128 and 8. Note that even for sample size is 8, it is 8X more expensive than BatchEnsemble in the testing time cost. Finally, we showed that combining BatchEnsemble and dropout ensemble leads to better accuracy and calibration. It is competitive to naive ensemble while keeping memory consumption efficient. It is also an evidence that BatchEnsemble is an orthogonal method to dropout ensemble; combining these two can potentially obtain better performance.

## 5 DIVERSITY ANALYSIS

As mentioned in Section 2, more diversity among ensembling members leads to better performance. Therefore, beyond accuracy and uncertainty metrics, we are particularly interested in how much diversity rank-1 perturbation provides. We compare BatchEnsemble to dropout ensemble and naive ensemble over the newly proposed diversity metric (Fort et al., 2019). The metric measures the disagreement among ensemble members on test set. We computed it over different amount of training data. See Appendix E for details on diversity metric and plots.

In this section, we give an intuitive explanation of why BatchEnsemble leads to more diverse members with fewer training data. If only limited training data is available, the parameters of the neural network would remain close to their initialization after convergence. In the extreme case where only one training data point is available, the optimization quickly converges and most of the parameters are not updated. This suggests that the diversity of initialization entirely determines the diversity of ensembling system. Naive ensemble has fully independent random initializations. BatchEnsemble has peudo-independent random initializations. In comparison, all ensemble members of dropout ensemble share the same initialized parameters. Therefore, both naive ensemble and BatchEnsemble significantly outperform dropout ensemble in diversity with limited training data.

More importantly, Figure 8 provides insightful advice on when BatchEnsemble achieves the best gain in practice. We observe that diversity of BatchEnsemble is comparable to naive ensemble when training data is limited. This explains why BatchEnsemble has higher gains on CIFAR-100 than CIFAR-10, because there are only 500 training points for each class on CIFAR-100 whereas 5000 on CIFAR-10. Thus, CIFAR-100 has more limited training data compared to CIFAR-10. Another implication is that BatchEnsemble can benefit more from heavily over-parameterized neural networks. The reason is that given the fixed amount of training data, increasing the number of parameters essentially converges to the case where the training data is limited. In practice, the best way to make full use of increasing computational power is to design deeper and wider neural networks. This suggests that BatchEnsemble benefits more from the development of computational power; because it has better gain on over-parameterized neural networks.

## 6 CONCLUSION

We introduced BatchEnsemble, an efficient method for ensembling and lifelong learning. BatchEnsemble can be used to improve the accuracy and uncertainty of any neural network like typical ensemble methods. More importantly, BatchEnsemble removes the computation and memory bottleneck of typical ensemble methods, enabling its successful application to not only faster ensembles but also lifelong learning on up to 100 tasks. We believe BatchEnsemble has great potential to improve in lifelong learning. Our work may serve as a starting point for a new research area.

### ACKNOWLEDGMENTS

YW was supported by University of Toronto Fellowship, Faculty of Arts And Science and Vector Scholarships in Artificial Intelligence (VSAI).

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

## A    DATASET DETAILS

**CIFAR:** We consider two CIFAR datasets, CIFAR-10 and CIFAR-100 (Krizhevsky, 2009). Each consists of a training set of size 50K and a test set of size 10K. They are natural images with 32x32 pixels. In our experiments, we follow the standard data pre-processing schemes including zero-padding with 4 pixels on each sise, random crop and horizon flip (Romero et al., 2015; Huang et al., 2016; Srivastava et al., 2015).

**WMT:** In machine translation tasks, we consider the standard training datasets WMT16 English-German and WMT14 English-French. WMT16 English-German dataset consists of roughly 4.5M sentence pairs. We follow the same pre-processing schemes in (Vaswani et al., 2017).Source and target tokens are processed into 37K shared sub-word units based on byte-pair encoding (BPE) (Britz et al., 2017). Newstest2013 and Newstest2014 are used as validation set and test set respectively. WMT14 English-French consists of a much larger dataset sized at 36M sentences pairs. We split the tokens into a 32K word-piece vocabulary (Wu et al., 2016).

**Split-CIFAR100:** The dataset has the same set of images as CIFAR-100 dataset (Krizhevsky, 2009). It randomly splits the entire dataset into T tasks so each task consists of $100/T$ classes of images. To leverage the task descriptor in the data, different final linear classifier is trained on top of feature extractor per task. This simplifies the task to be a $100/T$ class classification problem in each task. i.e. random prediction has accuracy $T/100$. Notice that since we are not under the setting of single epoch training, standard data pre-processing including padding, random crop and random horizontal flip are applied to the training set.

**Split-ImageNet:** The dataset has the same set of images as ImageNet dataset (Deng et al., 2009). It randomly splits the entire dataset into T tasks so each task consists of $1000/T$ classes of images. Same as Split-CIFAR100, each task has its own final linear classifier. Data preprocessing (He et al., 2016) is applied to the training data.

## B    IMPLEMENTATION DETAILS

In this section, we discuss some implementation details of BatchEnsemble.

**Weight Decay**: In the BatchEnsemble, the weight of each ensemble member is never explicitly calculated because we obtain the activations directly by computing Eqn. 5. To maintain the goal of no additional computational cost, we can instead regularize the mean weight $\overline{W}$ over ensemble members, which can be efficiently calculated as $\overline{W} = \frac{1}{B} W \circ S^\top R$, where $W$ is the shared weight among ensemble members, $S$ and $R$ are the matrices in Eqn. 5. We can also only regularize the

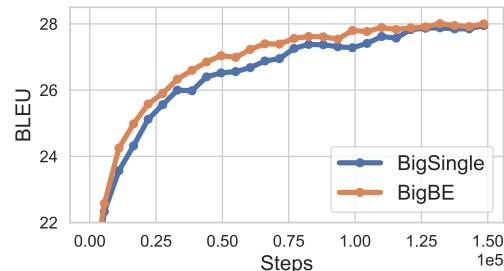

Figure 6: BLEU on English-German task.

shared weight and leave the fast weights unregularized because it only accounts for a small portion of model parameters. In practice, we find the above two schemes work equally.

**Diversity Encouragement**: Additional loss term such as KL divergence among ensemble members can be added to encourage diversity. However, we find it sufficient for BatchEnsemble to have desired diversity by initializing the fast weight ($s_i$ and $r_i$ in Eqn. 1) to be random sign vectors. Also note that the scheme that each ensemble member is trained with different sub-batch of input can encourage diversity as well. The diversity analysis is provided in Appendix G.

**Machine Translation**: The Transformer base is trained for 100K steps and the Transformer big is trained for 180K steps. The training steps of big model are shorter than Vaswani et al. (2017) because we terminate the training when it reaches the targeted perplexity on validation set. Experiments are run on 4 NVIDIA P100 GPUs. The BLEU score of Big Transformer on English-German task is in Figure 6. Although BatchEnsemble has lower perplexity as we showed in Section 4.2, we didn't observe a better BLEU score. Noted that the BLEU score in Figure 6 is lower than what Vaswani et al. (2017) reported. It is because in order to correctly evaluate model performance at a given timestep, we didn't use the averaging checkpoint trick. The dropout rate of Transformer base is 0.1 and 0.3 for

Transformer big on English-German while remaining 0.1 on English-French. For dropout ensemble, we ran a grid search between 0.05 and 0.3 in the testing time and report the best validation perplexity.

**Classification**: We train the model with mini-batch size 128. We also keep the standard learning rate schedule for ResNet. The learning rate decreases from 0.1 to 0.01, from 0.01 to 0.001 at halfway of training and 75% of training. The weight decay coefficient is set to be $10^{-4}$. We use an ensemble size of 4, which means each ensemble member receives 32 training examples if we maintain the mini-batch size of 128. It is because Batch Normalization (Ioffe & Szegedy, 2015) requires at least 32 examples to be effective on CIFAR dataset. As for the training budget, we train the single model for 250 epochs

## C    PREDICTIVE UNCERTAINTY

In this section, we evaluate the predictive uncertainty of BatchEnsemble on out-of-distribution tasks and ECE loss.

Similar to Lakshminarayanan et al. (2017), we first evaluate BatchEnsemble on out-of-distribution examples from unseen classes. It is known that deep neural network tends to make over-confident predictions even if the prediction is wrong or the input comes from unseen classes. Ensembles of models can give better uncertainty prediction when the test data is out of the distribution of training data. To measure the uncertainty on the prediction, we calculate the predictive entropy of Single neural network, naive ensemble and BatchEnsemble. The result is presented in Figure 7a. As we expected, single model produces over-confident predictions on unseen examples, whereas ensemble methods exhibit higher uncertainty on unseen classes, including both BatchEnsemble and naive ensemble. It suggests our ensemble weight generation mechanism doesn't degrade uncertainty modelling.

(a) Histogram of the predictive entropy on test examples from known classes, CIFAR-10 (left) and unknown classes, CIFAR-100 (right).

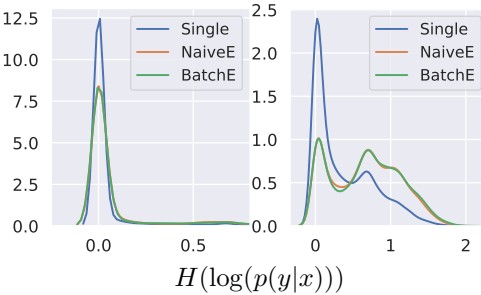

$$H(\log(p(y|x)))$$

(b) Expected Calibration Error. Ensemble of size 4. Lower ECE reflects better calibration.

|      | Single | MC-drop | BatchE | NaiveE |
|------|--------|---------|--------|--------|
| C10  | 3.27   | 2.89    | **2.37** | 2.32   |
| C100 | 9.28   | 8.99    | **8.89** | 6.82   |

We also calculate the Expected Calibration Error (Naeini et al., 2015) (ECE) of single model, naive ensemble and BatchEnsemble on both CIFAR-10 and CIFAR-100 in Table 7b. To calculate ECE, we group model predictions into M interval bins based on the predictive confidence (each bin has size $\frac{1}{M}$). Let $B_m$ denote the set of samples whose predictive probability falls into the interval $(\frac{m-1}{M}, \frac{m}{M}]$ for $m \in \{1, \ldots M\}$. Let $\text{acc}(B_m)$ and $\text{conf}(B_m)$ be the averaged accuracy and averaged confidence of the examples in the bin $B_m$. The ECE can de defined as the following,

$$\text{ECE} = \sum_{m=1}^{M} \frac{|B_m|}{n} |\text{acc}(B_m) - \text{conf}(B_m)| \tag{8}$$

where $n$ is the number of samples. ECE as a criteria of model calibration, measures the difference in expectation between confidence and accuracy (Guo et al., 2017). It shows that BatchEnsemble makes more calibrated prediction compared to single neural networks.

## D    UNCERTAINTY ON BANDITS

In this section, we conduct analysis beyond accuracy, where we show that BatchEnsemble can be used for uncertainty modelling in contextual bandits.

For uncertainty modelling, we evaluate our BatchEnsemble method on the recently proposed bandits benchmark (Riquelme et al., 2018). Bandit data comes from different empirical problems that highlight several aspects of decision making. No single algorithm can outperform every other

Table 4: Contextual bandits regret. Results are relative to the cumulative regret of the Uniform algorithm. We report the mean and standard error of the mean over 30 trials. Ensemble size with 4, 8. We remove the methods with mean rank greater than 10.

| | M.RANK | M.VALUE | MUSHROOM | STATLOG | FINANCIAL | JESTER | WHEEL |
|---|---|---|---|---|---|---|---|
| NaiveEnsemble4 | **5.30** | 34.64 | **13.44 ± 3.83** | **7.10 ± 1.15** | 11.31 ± 1.48 | 72.73 ± 6.32 | 68.63 ± 21.97 |
| NaiveEnsemble8 | 6.50 | 34.91 | 13.59 ± 3.13 | 7.15 ± 0.98 | 11.64 ± 1.57 | 73.54 ± 6.14 | 68.65 ± 19.32 |
| BatchEnsemble4 | 6.30 | 34.52 | 15.22 ± 5.21 | 11.53 ± 5.06 | 10.24 ± 2.66 | 72.65 ± 6.27 | 62.94 ± 26.12 |
| BatchEnsemble8 | 5.70 | **33.95** | 13.48 ± 3.36 | 9.85 ± 3.67 | 13.17 ± 2.87 | **71.84 ± 6.47** | 61.41 ± 26.18 |
| Dropout | 8.20 | 36.73 | 15.05 ± 8.23 | 9.31 ± 3.19 | 13.53 ± 2.98 | 71.90 ± 6.31 | 73.86 ± 22.48 |
| LinFullPost | 9.40 | 49.60 | 97.42 ± 4.52 | 19.00 ± 1.03 | 10.24 ± 0.92 | 78.40 ± 4.85 | **42.94 ± 12.68** |
| MultitaskGP | 5.90 | 34.59 | 12.87 ± 4.70 | 8.04 ± 3.77 | **8.50 ± 0.80** | 74.03 ± 5.96 | 69.52 ± 18.55 |
| RMS | 9.40 | 39.18 | 16.31 ± 6.13 | 10.44 ± 5.02 | 11.75 ± 2.64 | 73.38 ± 4.70 | 84.02 ± 24.67 |
| Uniform | 16.00 | 100.00 | 100.00 | 100.00 | 100.00 | 100.00 | 100.00 |

algorithm on every bandit problem. Thus, average performance of the algorithm over different problems is used to evaluate the quality of uncertainty estimation. The key factor to achieve good performance in contextual bandits is to learn a reliable uncertainty model. In our experiment, Thompson sampling samples from the policy given by one of the ensemble members. The fact that Dropout which is an implicit ensemble method achieves competitive performance on bandits problem suggests that ensemble can be used as uncertainty modelling. Indeed, Table 4 shows that BatchEnsemble with an ensemble size 8 achieves the best mean value on the bandits task. Both BatchEnsemble with ensemble size 4 and 8 outperform Dropout in terms of average performance. We also evaluate BatchEnsemble on CIFAR-10 corrupted dataset (Hendrycks & Dietterich, 2019) in Appendix C. Figure 5 shows that BatchEnsemble achieves promising accuracy, uncertainty and cost trade-off among all methods we compared. Moreover, combining BatchEnsemble and dropout ensemble leads to better uncertainty prediction.

# E DIVERSITY ANALYSIS

## E.1 DIVERSITY METRIC

Final performance metrics such as accuracy and uncertainty score we provided in Section 4 obscures many insights of our models. In this section, we provide visualization of some commonly used diversity metrics proposed in Fort et al. (2019). The diversity score used in the experiments below quantify the difference of two functions, by measuring fraction of the test data points on which their predictions disagree. This metric is 0 when two functions are making identical predictions, and 1 when they differ on every single example in the test set. We also normalize the diversity metric by the error rate to account for the case where random predictions provide the best diversity. There are other diversity metrics we can use such as the KL-divergence between the probability distributions. It doesn't make significant difference in our experiments, so we chose the fraction of disagreement for simplicity. We compare BatchEnsemble to dropout ensemble and naive ensemble. For BatchEnsemble and naive ensemble, we first select a model as our base model. We calculated the diversity measure of other ensembling members against the base model. We also plotted the diversity of the base model as a reference, which is trivially zero. For dropout ensemble, we sample a number of dropout masks and take the prediction of the first dropout mask as the base model. The disagreement fraction of the rest dropout masks can be computed against the base model.

In Figure 8, we plotted the diversity measures of BatchEnsemble, dropout ensemble and naive ensemble. According to the ensembling theory 2, the more diversity among ensembling members leads to better total accuracy. As Table 3 showed, naive ensemble achieves the best accuracy and then followed by BatchEnsemble and dropout ensemble. Therefore, we expect the diversity measure of BatchEnsemble is in the middle of naive ensemble and dropout ensemble. Figure 8a confirms our hypothesis. Note that naive ensemble, which only differs from random initializations, is very effective at sampling diverse and accurate solutions. This is aligned to the observation in Fort et al. (2019). Notice that naive ensemble incurs 4X more memory cost and 3X more inference time than Rank-1 Net. Thus, we can conclude that BatchEnsemble achieves the best trade-off among accuracy, diversity and efficiency in all methods we compared. The diversity gap between naive ensemble and BatchEnsemble is due to the limited expressiveness of rank-1 perturbation. This provides scope for future research direction.

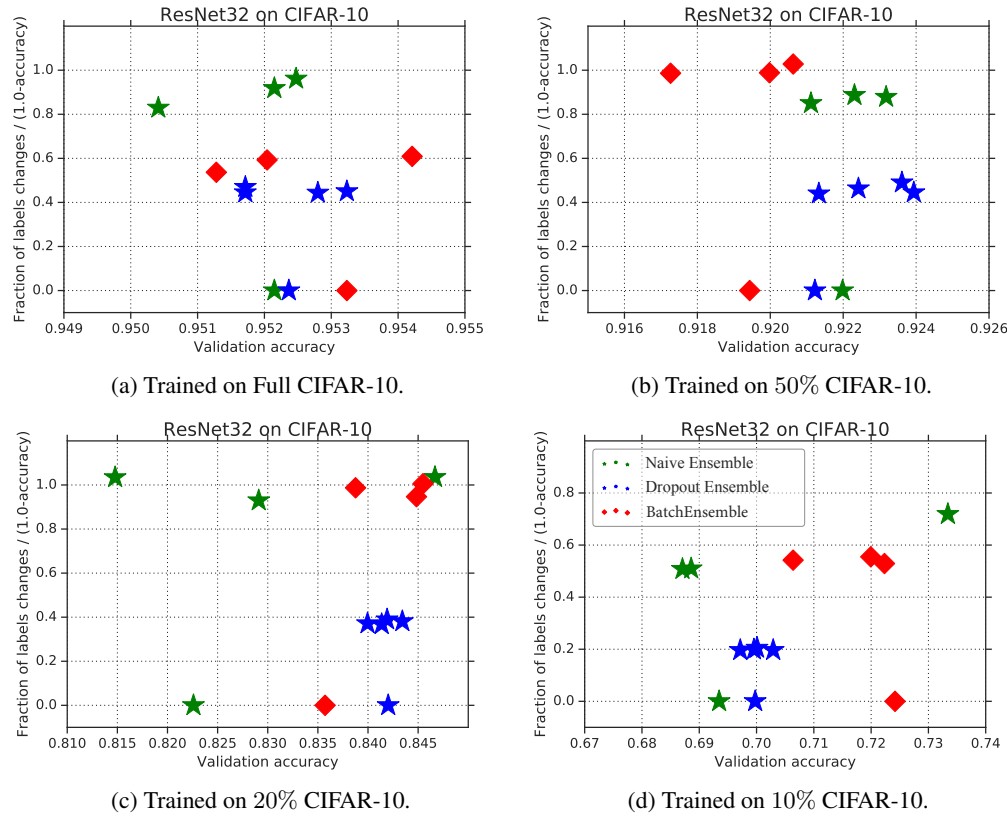

Figure 8: Comparison among BatchEnsemble, naive ensemble and dropout ensemble over diversity metric. Each point in the plot represents a trained model where x-axis represents its accuracy on validation set and y-axis represents its diversity against the base model. The base model trivially has 0 diversity. We plot the diversity of models trained on different proportions of training data, respectively 100%, 50%, 20% and 10%.

## E.2 DIVERSITY METRIC ON PARTIAL TRAINING SET

There are two sources of uncertainty: *aleatoric* uncertainty and *epistemic* uncertainty. *Epistemic* uncertainty accounts for the uncertainty in the model we train. We focus more on *epistemic* uncertainty because the *aleatoric* uncertainty is difficult to measure. The simplest to magnify *epistemic* uncertainty is to reduce the number of training data points. Under this case, we want our model to be more uncertain to reflect the lack of training data. Ovadia et al. (2019) showed that more diversity among ensembling member leads to larger uncertainty score. There, under the case where only limited training data is provided, we hope that our ensembling method can produce more diverse member, compared to training on full dataset.

We repeated the experiments in Appendix E.1 with the exactly same diversity metric on models trained with proportional CIFAR-10 dataset. Figure 8b, Figure 8c and Figure 8d plotted the diversity measure with 50%, 20%, and 10% CIFAR-10 training data, respectively. The results showed that BatchEnsemble is on par with naive ensemble under limited training data. In Figure 8b, it shows that when we reduce the number of training data to a half, BatchEnsemble achieves comparable diversity to naive ensemble. It outperforms dropout ensemble by a significant margin. Figure 8c and Figure 8d confirm the conclusion under even fewer training data points. Given the fact that diversity reflects *epistemic* uncertainty, we want our model to have more diversity when only limited training data is available. However, Figure 8 showed that dropout ensemble has the same diversity on 100%, 50%, and 20% training data. This is a significant flaw of dropout ensemble. In conclusion, BatchEnsemble leads to much more diverse member than dropout ensemble under the case of limited training data. To supplement Figure 8, we provide the ensembling accuracy of BatchEnsemble, naive ensemble and dropout ensemble trained on proportional training set in Table 5.

Table 5: Validation accuracy on ResNet32 with proportional training data. Ensemble with size 4. MC-drop stands for dropout ensemble (Gal & Ghahramani, 2015). Single represents the accuracy of the base model in naive ensemble in Figure 8.

|  | Single | MC-drop | BatchEnsemble | NaiveEnsemble |
|---|---|---|---|---|
| CIFAR-10 | 95.22 | 95.40 | **95.61** | 96.09 |
| CIFAR-10 (50%) | 92.20 | 92.43 | **92.93** | 93.01 |
| CIFAR-10 (20%) | 82.32 | 84.5 | **86.08** | 86.17 |
| CIFAR-10 (10%) | 69.37 | 71.3 | **76.75** | 76.77 |

## F    COMPARISON TO NAIVE ENSEMBLE OF SMALL MODELS

In this section, we compare BatchEnsemble to naive ensemble of small models on CIFAR-10/100 dataset. To maintain the same memory consumption as BatchEnsemble, we trained 4 independent ResNet14x4 models and evaluate the naive ensemble on these 4 models. This setup of naive ensemble still has roughly $10\%$ memory overhead to BatchEnsemble. The results are reported in Table 6. It shows that naive ensemble of small models achieves lower accuracy than BatchEnsemble. It illustrates that given the same memory budget, BatchEnsemble is a better choice over naive ensemble.

Table 6: Supplementary result to Table 3. NaiveSmall is naive ensemble of 4 ResNet14x4 models. Vanilla, MC-drop and BatchEnsemble are still ResNet32x4 as in Table 3.

|  | Vanilla | MC-drop | BatchEnsemble | NaiveSmall |
|---|---|---|---|---|
| CIFAR10 | 95.31 | 95.72 | **95.94** | 95.59 |
| CIFAR100 | 78.32 | 78.89 | **80.32** | 79.09 |

## G    PREDICTIVE DIVERSITY

As we discussed in Section 2, ensemble benefits from the diversity among its members. We focus on the set of test examples on CIFAR-10 where single model makes confident incorrect predictions while ensemble model predicts correctly. We used the final models we reported in Section 4.3. In Figure 9, we randomly select examples from the above set and plot the prediction map of single model, each ensemble member and mean ensemble. As we can see, although some of the ensemble members make mistakes on thoes examples, the mean prediction takes the advantage of the model averaging and achieves better accuracy on CIFAR-10 classification task. We notice that BatchEnsemble preserves the diversity among ensemble members as naive ensemble.

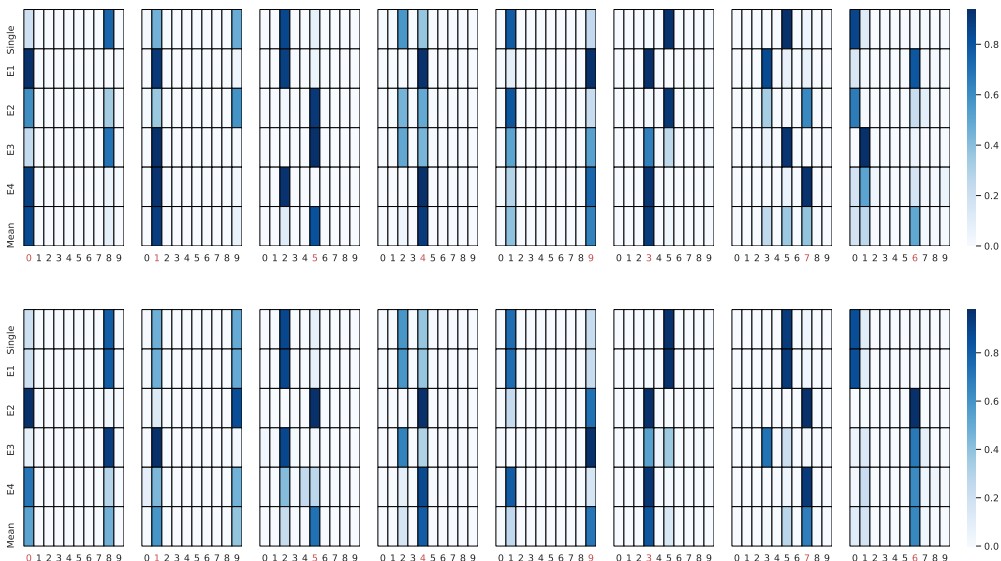

Figure 9: Visualizing prediction diversity among BatchEnsemble (top row) and naive ensemble (bottom row) members on selected test examples on CIFAR-10. The y-axis label denotes mean prediction of ensemble (Mean), individual ensemble member prediction (from E1 to E4) and single model prediction (Single). Correct class is labelled as red. BatchEnsemble preserves the model diversity as naive ensemble.

