# OpenReview forum: "BatchEnsemble: an Alternative Approach to Efficient Ensemble and Lifelong Learning"
_ICLR.cc/2020/Conference — Accept (Poster)_

### Official Review · AnonReviewer1 · 2019-10-22
**Official Blind Review #1**

**Rating:** 3

**Review:**

This paper aims to improve the efficiency of ensembles of neural nets in traditional supervised learning and life-long learning (learning on a series of tasks). The main idea is to let all the neural nets in an ensemble share the same weights W for each layer, and the weights for each neural net is generated by the Hadamard product of W and a specific rank-one matrix of the same size as W that is different across members in the ensemble. In experiments, they evaluate the method with some baselines on life-long learning, traditional classification, NMT tasks, and uncertainty modeling.

The paper relates the proposed method to several different learning problems and applications and lists many potential advantages in these applications: it covers a lot of things. However, it lacks in-depth discussion to several key problems, rigorous analysis or complete experimental study to support the main claims, for example:

Why can the simple method achieve a more compelling trade-off between accuracy and efficiency/memory costs comparing to a large single model or a naive ensemble of small models? Any mathematical or information-theoretical explanation behind that?

It is easy to understand that the ensemble defined here can improve efficiency and reduce memory cost. But as an alternative to the naive ensemble, we also expect the performance to not suffer from severe drawbacks. How to control efficiency-performance trade-off in the proposed method?

How were the baselines for each experiment selected? How to determine the specific setting in each experiment (any reason behind choosing the parameters in the settings)?

In the life-long learning settings, the shared weights W is only trained on the first task and then keeps fixed: this can leads to both large variance and bias. Why does it simply work well without causing any serious problems?

The rank-one extension of a shared model W enforces a very strong regularization to the model for each task. Will the method work promisingly when the tasks are more different from each other or harder to solve? For example, what if we increase the classes in each task? Is the rank-one extension still flexible and expressive enough to handle this situation?

These are some of the most important questions needed to be answered in the first place before showing higher evaluation metrics and listing the potential advantages of the proposed method. But it is not clear to me at all how they can be answered according to the contents in the current paper. I notice that the authors mentioned the last two questions at the end of Section 3, but no explanations/discussions were given.

Other major concerns:

1) Mathematically, comparing to single model Wx, the proposed ensemble method equals to applying a dimension-wise scaling to the input x and a dimension-wise scaling to the output Wx, and the scaling factors vary across different tasks. Hence, the proposed structure is exactly the same as fine-tuning two groups of batch normalization scaling factors before and after applying transformation W. It does not make much sense in the experiments that the performance of BN-Tuned in Figure 3a is much worse than the proposed method since they share exactly the same structure and math (note the memory and computational costs are also the same). The paper does not give an explanation about this. Moreover, the baseline BN-Tuned is only compared on only one of those datasets in the paper. It should be one of the most important baselines and needs to be compared in all experiments.

2) On each benchmark dataset (except the last one), only 1-2 baselines are compared and most baselines are not state-of-the-art methods or not methods specifically designed for the problem (e.g., many are dropout and its variants). This makes the comparisons not convincing, especially considering that the experimental settings are determined by the authors and might be chosen for the best performance of the proposed method.

3) At least two baselines should be included in all experiments: 1) single model with the equal number of model parameters, and 2) naive ensemble not sharing parameters across member models. However, each experiment only includes one or even none of these two baselines.

4) Memory and training/test computational costs need to be reported for each experiment. However, the currently reported results are incomplete here and there.

5) Comparing to the currently limited number of baselines on the incomplete evaluation metrics, the proposed method does not show significant improvements, for example, the results in Figure 4, Table 1 and Table 2.

6) The proposed method requires the models for different tasks should have exactly the same architecture. This could be a strong limitation in many scenarios. For example, when different tasks have significantly different numbers of classes.

**Experience Assessment:**

I have published one or two papers in this area.

**Review Assessment: Checking Correctness Of Derivations And Theory:**

I carefully checked the derivations and theory.

**Review Assessment: Checking Correctness Of Experiments:**

I carefully checked the experiments.

**Review Assessment: Thoroughness In Paper Reading:**

I read the paper thoroughly.

---

> ### Author Response · Authors · 2019-11-09
> **Response to reviewer #1 (1)**
>
> Thank you for your careful and insightful feedback. We first answer some questions without extra experiments.
>
> -> Q: Why can the simple method achieve a more compelling trade-off between accuracy and efficiency/memory costs comparing to a large single model or a naive ensemble of small models? Any mathematical or information-theoretical explanation behind that?
>
> Combining rank-1 perturbation per layer with large number of layers in deep networks leads to diverse ensemble member as evidenced by Figure 8 in the Appendix. In ensemble theory, more diversity leads to better accuracy by averaging the output. Diversity is also the key to better uncertainty prediction. Thus, the uncertainty modelling experiments are another evidence that rank-1 perturbation provides satisfying diversity. This explains why rank-1 perturbation achieves an improved accuracy.
>
> Regarding a theoretical explanation, our hypothesis is that a low-dimensional subspace of the parameters (particularly restricted to the first rank of each weight) provides sufficient expressivity and trainability to return diverse solutions. [1] shows that we can construct the subspace by taking the first principal components of SGD trajectory, which leads to diverse sets of high performing models ($w=w_0 + Pz$ where $w_0$ is a solution point found by SGD, z is the vector from the constructed subspace, P is a linear transformation). The slow weight in our paper can be thought of w_0 and the fast weights are vectors in the subspace.
>
> There are other related works adding evidence to this hypothesis: for predictive accuracy, the intrinsic dimensionality of networks has been signalled to be very low [2]; and for predictive uncertainty, a concurrent ICLR submission (https://openreview.net/forum?id=BkxREeHKPS) shows that tying uncertainty parameters up to a certain low rank may be sufficient.
>
> [1]: Izmailov, P., Maddox, W.J., Kirichenko, P., Garipov, T., Vetrov, D.P., & Wilson, A.G. (2019). Subspace Inference for Bayesian Deep Learning. ArXiv, abs/1907.07504.
> [2]: Li, C., Farkhoor, H., Liu, R., & Yosinski, J. (2018). Measuring the Intrinsic Dimension of Objective Landscapes. ArXiv, abs/1804.08838.
>
> Overall, there are two related questions here:
> 1. Does rank-1 perturbation provide diversity?
> 2. How much diversity does it provide?
> In this paper, we focus on answering the first question. We leave formally understanding how and why BatchEnsemble provides a compelling accuracy-efficiency tradeoff to future work. Understanding diversity of naive ensembles is already an open research direction (the work was published in 2016; one work focused solely on the analysis is a concurrent ICLR submission, https://openreview.net/forum?id=r1xZAkrFPr).”
>
> BatcheEnsemble v.s. A large single model: In the experiment section, we compared BatchEnsemble to single model with roughly the same number of parameters (except the machine translation part). More specifically, BatchEnsemble occurs 2% parameters overhead. With such parameters budget, we can only scale the number of filters in single model by 1.07, which results to no improvement. BatchEnsemble can also be used as uncertainty modelling and lifelong learning which the single model is not capable of.
>
> BatchEnsemble v.s. Naive ensemble of small models: If the ensemble size is 4 then the fair comparison is (naive ensemble of 4 ResNet-14 vs. BatchEnsemble of ResNet-32). We think this is a good suggestion and will add the comparison in the revision. But BatchEnsemble still has better testing time cost. And if ensemble size is large such as Split-ImageNet, BatchEnsemble is still a better choice than naive ensemble of small models.
>
> -> Q: It is easy to understand that the ensemble defined here can improve efficiency and reduce memory cost. But as an alternative to the naive ensemble, we also expect the performance to not suffer from severe drawbacks. How to control efficiency-performance trade-off in the proposed method?
>
> We propose BatchEnsemble as an alternative efficient ensemble method. Rather than comparing to naive ensemble (the memory consumption is 2 v.s. 400 with ensemble size 4), a more fair comparison is comparing to dropout-ensemble which we consider in 4.3 and 4.4 and Appendix-D.
>
> In general, naive ensembles, BatchEnsemble, and dropout ensembles are different points on the tradeoff curve of efficiency-to-performance. Ensembles are not concerned with efficiency (Figure 1), but achieve highest performance. If one prefers high efficiency and wants to find the method with highest performance, then we show over many tasks that BatchEnsemble is a better alternative to dropout (and single models). BatchEnsemble can even perform as well as naive ensembles on some tasks, but they are not meant to be an alternative if efficiency is not a concern.

---

> > ### Author Response · Authors · 2019-11-09
> > **Response to reviewer #1 (2)**
> >
> > -> Q: How were the baselines for each experiment selected? How to determine the specific setting in each experiment (any reason behind choosing the parameters in the settings)?
> >
> > How baseline selected in lifelong learning experiment is explained in the second and fourth paragraph in section 4.1. The specific setting in the experiments followed exactly as [1,2], except the single-epoch setting in their papers. For other experiments, we compare to single model, naive ensemble and dropout-ensemble except the machine translation experiment. The experiments setting are commonly used in other papers.
> >
> > -> Q: In the life-long learning settings, the shared weights W is only trained on the first task and then keeps fixed: this can leads to both large variance and bias. Why does it simply work well without causing any serious problems? The rank-one extension of a shared model W enforces a very strong regularization to the model for each task. Will the method work promisingly when the tasks are more different from each other or harder to solve? For example, what if we increase the classes in each task? Is the rank-one extension still flexible and expressive enough to handle this situation?
> >
> > The rank-1 perturbation per layer provides satisfying expressiveness because we chose deep neural network (at least 32 layers) in our experiments. The gap between BatchEnsemble and PNN increases to 3% (70.1 v.s 73.0) if we use AlexNet. If we increase the number of classes in each task, all methods would have decreasing accuracy. So in Split-CIFAR100, we chose the set-up the same as (GEM, AGEM paper). In Split-ImageNet, we chose T=100 to demonstrate that BatchEnsemble is capable of learning a large number of sequential tasks.
> > We plan to combine the improvement we mentioned in section 3.3 with harder lifelong learning tasks (such as learning sequential skills in RL) in future work. Note that Split-CIFAR100 is already a challenging domain for which state-of-the-art methods evaluate on [1,2,3].
> >
> > [1]: Lopez-Paz, D., & Ranzato, M. (2017). Gradient Episodic Memory for Continual Learning. NIPS.
> > [2]: Chaudhry, A., Ranzato, M., Rohrbach, M., & Elhoseiny, M. (2018). Efficient Lifelong Learning with A-GEM. ArXiv, abs/1812.00420.
> > [3]: Xu, J., & Zhu, Z. (2018). Reinforced Continual Learning. NeurIPS.
> >
> > -> Q: Mathematically, comparing to single model Wx, the proposed ensemble method equals to applying a dimension-wise scaling to the input x and a dimension-wise scaling to the output Wx, and the scaling factors vary across different tasks. Hence, the proposed structure is exactly the same as fine-tuning two groups of batch normalization scaling factors before and after applying transformation W. It does not make much sense in the experiments that the performance of BN-Tuned in Figure 3a is much worse than the proposed method since they share exactly the same structure and math (note the memory and computational costs are also the same). The paper does not give an explanation about this. Moreover, the baseline BN-Tuned is only compared on only one of those datasets in the paper. It should be one of the most important baselines and needs to be compared in all experiments.
> >
> > That’s a very interesting connection. Note that while the connection is true (i.e., apply BN at every input and pre-activation, but remove shift parameter and batch-statistic normalization), the intuition behind why that works is not really clear from the BN perspective. Our work interprets feature-wise scaling of inputs and preactivations as a rank-1 perturbation of the weights, which is meant to provide expressivity. It’s uncommon to think of BN’s feature scaling as still BN if the computation doesn’t use batch statistics. Additionally, our method can perturb each dimension of the 4-d weights in convolutional layers while the BN interpretation can’t.
> >
> > -> Q: On each benchmark dataset (except the last one), only 1-2 baselines are compared and most baselines are not state-of-the-art methods or not methods specifically designed for the problem (e.g., many are dropout and its variants). This makes the comparisons not convincing, especially considering that the experimental settings are determined by the authors and might be chosen for the best performance of the proposed method.
> >
> > Comparing to dropout-ensemble because it is the state-of-the-art memory efficient ensemble as far as we know. The experimental setting is chosen to be the same as previous published papers.
> >
> > -> Q: At least two baselines should be included in all experiments: 1) single model with the equal number of model parameters, and 2) naive ensemble not sharing parameters across member models. However, each experiment only includes one or even none of these two baselines.
> >
> > The only experiment that doesn’t have naive ensemble is machine translation. It is an upper bound so we don’t think missing it in one of many experiments should affect the quality of the paper. We plan to add it in the revision.

---

> > > ### Author Response · Authors · 2019-11-09
> > > **Response to reviewer #1 (3)**
> > >
> > > -> Q: Memory and training/test computational costs need to be reported for each experiment. However, the currently reported results are incomplete here and there.
> > >
> > > Memory and training/test computational costs (relative to single model) are consistent across experiments. We showed the cost in Figure 1. A comparison to other continual learning methods are given in Table 4 in the Appendix. We will state the costs in a more clear way in each experiment in the revision.
> > >
> > > -> Q: Comparing to the currently limited number of baselines on the incomplete evaluation metrics, the proposed method does not show significant improvements, for example, the results in Figure 4, Table 1 and Table 2.
> > >
> > > It is not fair to compare BatchEnsemble to naive ensemble given the memory cost. We show a significant improvement for low memory cost tradeoff (i.e., significant improvement over single model and dropout-ensemble).
> > >
> > > -> Q: The proposed method requires the models for different tasks should have exactly the same architecture. This could be a strong limitation in many scenarios. For example, when different tasks have significantly different numbers of classes.
> > >
> > > BatchEnsemble can ensemble of network with different length. It is true that it has some limitations and is a potential for future work. Note that other memory efficient methods such as dropout-ensemble have the same limitation; and existing SOTA for lifelong learning like GEM, A-GEM also have this restriction.

---

> ### Author Response · Authors · 2019-11-15
> **More on the two groups of BN interpretation.**
>
> Besides the argument in the initial response, we want to add in the lifelong learning experiments, we don't need to vectorize the BatchEnsemble computation because only one set of fast weight is involved in each task. Thus, our framework can naturally extend to higher rank perturbation which adds more expressiveness to our model although we found it is not necessary for the lifelong learning task in this paper. Such perturbation cannot be interpreted as two groups of BN (one of them has no BN statistics).

---

### Official Review · AnonReviewer3 · 2019-10-22
**Official Blind Review #3**

**Rating:** 6

**Review:**

This paper presents an ensemble method for neural networks, named BatchEnsemble, that aims to provide the benefits of improved accuracy and predictive uncertainty of traditional ensembles but with a significantly lower computational cost and memory cost. The method works by maintaining a shared “slow” weight matrix per layer, along with an ensemble of rank-1 “fast” weight matrices that are combined individually with the slow matrix via a Hadamard product in order to generate the network ensemble. The fast matrices can be stored as a pair of vectors, incurring a much smaller memory cost than a full rank matrix, and the prediction of an ensemble member can be vectorized such that the forward pass through the whole ensemble can be parallelized within a single GPU, yielding a computational speedup over traditional ensembles. The method is evaluated across a host of experimental settings, including image classification, machine translation, lifelong learning and uncertainty modelling.
Overall, I recommend this paper to be accepted because:
(i) the method proposed is simple to understand and implement,
(ii) it yields clear computation and memory benefits over a traditional ensemble,
(iii) the method is motivated by a good literature review, putting the approach and experiments conducted in context,
(iv) while in terms of performance the experimental results are mixed, many different settings are evaluated, they are conducted fairly and they are transparently described, with the limitations are clearly acknowledged for the most part.

Specific comments / questions
* Issues with BatchEnsemble as a lifelong learning method. When applied to lifelong learning, the slow weights are only tuned for the first task - as acknowledged by the authors, this means that forward transfer is only possible from the 1st task to all subsequent tasks and, more concerningly, it could severely limit the expressiveness of the ensemble for subsequent tasks that can only make a rank-1 adjustment to each layer. On the split-cifar and split-imagenet tasks this interestingly does not seem to be an issue, but one could imagine that it could be for tasks that differ more.
    * Was the task split and order randomised for each run in Figure 3a and 3b? Would be interesting to know if the choice of first task matter for performance. Also, did the authors try not training the slow weights at all for the lifelong learning experiments? This would show how much the transfer from the first task helps the subsequent ones.
    * In Figure 3b, it’s strange that EWC has a similar/ slightly higher forgetting than a vanilla neural network - do the authors have an explanation for this? Was the regularisation coefficient tuned for EWC?
    * The proposed solution for enabling transfer beyond the 1st task is to enable lateral weights from features from previous tasks, as in progressive neural networks, but this would undermine the parameter efficiency of the model.
* Machine translation experiments.
    * BatchEnsemble on the attention layers of a transformer speeds up training in machine translation, but has little effect on final performance of the model versus a single transformer.
    * Were any measures taken to equalise the number of parameters in the single transformer versus the ensemble method?
    * Was a naive ensemble trained on the machine translation tasks for comparison?
* Image classification experiments.
    * It is hard to fairly compare the BatchEnsemble performance to the single model performance here given the 50% extra training iterations, but its encouraging that BatchEnsemble outperforms MC-dropout and comes close to the performance of a naive ensemble.
* Predictive uncertainty / diversity. BatchEnsemble seems to perform well for uncertainty modelling in contextual bandits relative to a number of baselines.
* How can the method be used as a basis for future work? It would be good to see some discussion of whether and how BatchEnsemble could be combined with other neural network ensemble methods.

Minor comments not affecting review:
* Section 4.4, paragraph 2, line 1 “uncertainty” misspelt.

**Experience Assessment:**

I have read many papers in this area.

**Review Assessment: Checking Correctness Of Derivations And Theory:**

I assessed the sensibility of the derivations and theory.

**Review Assessment: Checking Correctness Of Experiments:**

I assessed the sensibility of the experiments.

**Review Assessment: Thoroughness In Paper Reading:**

I read the paper thoroughly.

---

> ### Author Response · Authors · 2019-11-09
> **Response to reviewer #3**
>
> Thank you for your careful and insightful feedback. We first answer some questions without extra experiments.
>
> -> Q: Issues with BatchEnsemble as a lifelong learning method. When applied to lifelong learning, the slow weights are only tuned for the first task - as acknowledged by the authors, this means that forward transfer is only possible from the 1st task to all subsequent tasks and, more concerningly, it could severely limit the expressiveness of the ensemble for subsequent tasks that can only make a rank-1 adjustment to each layer. On the split-cifar and split-imagenet tasks this interestingly does not seem to be an issue, but one could imagine that it could be for tasks that differ more.
>
> We have a potential solution to make forward transfer possible as mentioned in section 3.3. On the benchmark we consider (Split-CIFAR100 is already considered a hard lifelong learning dataset), BatchEnsemble shows promising performance. We’re not aware of existing challenging lifelong learning datasets that necessitate a more complicated solution. So we plan to implement the potential solution in harder lifelong learning tasks as future work.
>
> -> Q: Was the task split and order randomised for each run in Figure 3a and 3b? Would be interesting to know if the choice of first task matter for performance. Also, did the authors try not training the slow weights at all for the lifelong learning experiments? This would show how much the transfer from the first task helps the subsequent ones.
>
> It is randomised and the choice of first task affects the performance. The experiments are running with 5 random seeds and the confidence interval is plotted in Figure 3. We tried not to train the slow weights (just train the fast weights) for the first task. The model severely underfits so learning a reasonable slow weight is crucial for better performance in lifelong learning.
>
> -> Q: In Figure 3b, it’s strange that EWC has a similar/ slightly higher forgetting than a vanilla neural network - do the authors have an explanation for this? Was the regularisation coefficient tuned for EWC?
>
> Answer: A typo in the forgetting measure (it should be 0.12 for PNN). We will fix it in the revision.
>
> -> Q: The proposed solution for enabling transfer beyond the 1st task is to enable lateral weights from features from previous tasks, as in progressive neural networks, but this would undermine the parameter efficiency of the model.
>
> We agree that the naive implementation of lateral weights would undermine the parameter efficiency although it is still more efficient than PNN. Assuming some similarity among tasks, maybe we just need sparse lateral connection to previous tasks and maintain parameter efficiency. It is an interesting direction for future work.
>
> -> Q: Machine translation experiments: BatchEnsemble on the attention layers of a transformer speeds up training in machine translation, but has little effect on final performance of the model versus a single transformer. Were any measures taken to equalise the number of parameters in the single transformer versus the ensemble method? Was a naive ensemble trained on the machine translation tasks for comparison?
>
>
> We showed some improvement on perplexity over single model. However, with multiple ensemble members, it has the potential to make calibrated predictions in machine translation. There is no standard uncertainty benchmark in machine translation so we didn’t include it in the experiment section. We slightly increased the number of units in the FC (given only 2% extra parameters budget) which has no gain over single model. We are running naive ensemble experiments now and plan to add it in the revision.
>
> -> Q:  Image classification experiments: It is hard to fairly compare the BatchEnsemble performance to the single model performance here given the 50% extra training iterations, but its encouraging that BatchEnsemble outperforms MC-dropout and comes close to the performance of a naive ensemble.
>
> Increasing the training iterations for single model has no further improvement. If we increase the batch size (which takes more advantage of modern hardware), we don’t increase the number of training iterations.
>
> -> Q: How can the method be used as a basis for future work? It would be good to see some discussion of whether and how BatchEnsemble could be combined with other neural network ensemble methods.
>
> We believe BatchEnsemble is orthogonal to other ensemble methods such as SWA, Snapshot ensemble, dropout-ensemble. One can potentially combine these methods with BatchEnsemble. We will add more discussion in the revision. We also have experiments results that combining BatchEnsemble and Dropout leads to better uncertainty modelling, which will be added in the revision.

---

> > ### Comment · AnonReviewer3 · 2019-11-10
> > **Satisfied with responses - keeping score**
> >
> > Thank you for your replies to my questions. I'm satisfied overall and keep my weak accept score - my main concerns are still (i) the limitations of the method for lifelong learning when extended to more complex/unrelated tasks and (ii) the mixed results vs the baselines in the experiments conducted.

---

> > > ### Author Response · Authors · 2019-11-13
> > > **Combining BatchEnsemble with dropout leads to better calibration.**
> > >
> > > We would like to thank the reviewer for the positive feedback. We add one more experiment on CIFAR-10 corruption dataset in the appendix D in revision. Figure 7 shows that BatchEnsemble achieves the best trade-off among memory, testing cost, accuracy and calibration. Moreover, it shows that combining BatchEnsemble and dropout leads to even better performance. It is even competitive to naive ensemble while maintaining single model memory cost. It is an evidence that BatchEnsemble is orthogonal to current ensemble method such as dropout ensemble. It also answers the question that how the method can be used as a basis for future work.
> > >
> > > About the two main concerns:
> > > (i) Split-CIFAR100 is considered as a challenging lifelong learning task in previous published papers. We also extend our method to Split-ImageNet which we think is even harder because of the large number of sequential tasks. Our method is an inspiration for applying memory efficient ensemble in lifelong learning task. We think it is fair to leave [(i) developing harder lifelong learning benchmark (ii) extending our framework to it] to future work.
> > > (ii) BatchEnsemble achieves better performance than single model and dropout ensemble over all experiments we showed (except the machine translation one where we are still working on). Given the tidy memory overhead BatchEnsemble introduces, we don't think it is fair to compare BatchEnsemble to naive ensemble which is supposed to be an upper bound for all memory efficient methods.

---

### Official Review · AnonReviewer2 · 2019-10-23
**Official Blind Review #2**

**Rating:** 6

**Review:**

The paper proposes a new efficient ensembling method that has smaller memory footprint than naive ensembling and allows a simple parallelization on one device. The authors’ idea is based on sharing weights between individual ensembling models. The weights of each model can be represented as element-wise product of two matrices: shared one and matrix with rank 1 that can be efficiently stored.

The idea is quite interesting despite its simplicity. The experimental part is quite broad. I would like to highlight the lifelong learning as the strongest experimental result achieved by the authors. Despite the significant improvement on top of the baselines, this approach has one drawback described by the authors themselves. This method is difficult to generalize for the case of very diverse tasks despite its scalability. Nevertheless, I would not consider it as a large problem.

I have a concern regarding ensembling. Do I understand correctly that in Figure 1 the method achieves almost constant test time cost only in the case of one device parallelization? If yes, then Figure 1 is slightly misleading and the description of this figure should be improved.
In the classification section the authors compare their approach only with MC-dropout. I would recommend adding other ensembling methods that have small memory footprint: e.g. [1], and can be better than MC-dropout. The same is true for machine translation section.

The authors emphasize that their approach is faster than consequently training independent models. However,  since these models are independent, it is possible to train them in parallel on multiple devices. The restriction to one device during training seems in general a bit artificial.


[1]  Stefan  Lee,  Senthil  Purushwalkam,  Michael  Cogswell,  David  Crandall,  and  Dhruv  Batra.   Whym  heads  are  better  than  one:  Training  a  diverse  ensemble  of  deep  networks.arXiv  preprintarXiv:1511.06314, 2015b


Overall, it is an interesting paper, that has several drawbacks.


**Experience Assessment:**

I have published one or two papers in this area.

**Review Assessment: Checking Correctness Of Derivations And Theory:**

I assessed the sensibility of the derivations and theory.

**Review Assessment: Checking Correctness Of Experiments:**

I assessed the sensibility of the experiments.

**Review Assessment: Thoroughness In Paper Reading:**

I read the paper at least twice and used my best judgement in assessing the paper.

---

> ### Author Response · Authors · 2019-11-09
> **Response to reviewer #2**
>
> Thank you for your careful and insightful feedback.
>
> -> Q: I have a concern regarding ensembling. Do I understand correctly that in Figure 1 the method achieves almost constant test time cost only in the case of one device parallelization? If yes, then Figure 1 is slightly misleading and the description of this figure should be improved.
>
> Figure 1 is supposed to help understanding BatchEnsemble in the matrix element-wise multiplication view. For efficient computation, we use the vectorized computation. We will make this clear in the revision.
>
> -> Q: In the classification section the authors compare their approach only with MC-dropout. I would recommend adding other ensembling methods that have small memory footprint, and can be better than MC-dropout. The same is true for machine translation section.
>
> TreeNet is an ensemble method that is more memory expensive than MC-dropout and BatchEnsemble. Which layers should be shared among ensemble members in deep network such as ResNet-32 is still unknown and need extra effort to discover. We will include this work in discussion in the revision and run experiments that compares to it if time permitted.
>
> -> Q: The authors emphasize that their approach is faster than consequently training independent models. However,  since these models are independent, it is possible to train them in parallel on multiple devices. The restriction to one device during training seems in general a bit artificial.
>
> We don’t restrict to one device (in fact, experiments (4.1, 4.2) use multiple devices). Rather, we show that BatchEnsemble can exploit parallel training using both axes (within a device as well as across devices) and this depends on the user. In the extreme setting benefiting naive ensembles where all parallelism is done across devices, note that the memory cost for parameters still remains smaller for BatchEnsemble (assuming a parameter server).

---

### Author Response · Authors · 2019-11-15
**Summary of revision, including new experiments**

The revision includes:
1. An additional experiments about neural network calibration on CIFAR-10 corruption dataset in Appendix D. Figure 7 shows that BatchEnsemble achieves the best trade-off among accuracy, calibration, computational and memory costs. Moreover, it shows that BatchEnsemble is orthogonal to existing ensemble methods such as Dropout ensemble. Combining dropout ensemble and BatchEnsemble leads to better calibration (as pointed out by reviewer #3). It is even competitive to naive ensemble while has 4x less memory cost.

2. We added the result of MC-drop on Transformer in Table 1. Transformer single model already heavily uses dropout as regularization so dropout ensemble doesn't lead to better perplexity during testing. We didn't have time to finish the naive ensemble experiments but we don't think this degrades the quality because naive ensemble is supposed to be an upper bound. We have naive ensemble in every other experiments if it is feasible.

3. As reviewer # 1 asked, we compared BatchEnsemble to naive ensemble of small models so that they share the same memory budget in Table 5, Appendix F.

4. We fixed a number of typos.

---

### Public Comment · ~Ziyu_Wang2 · 2020-02-11
**Related work on scalable ensembles for uncertainty quantification**

Dear authors,

Congratulations on having your paper accepted! It's very nice to see uncertainty-aware learning methods improving performance on modern DL tasks, and the application on continual learning is particularly interesting.

In the following work we have also studied ensemble-like method for uncertainty estimation, with a similar focus of maintaining the training scheme for existing network architectures. I hope you will find the reference helpful.

Z. Wang, T. Ren, J. Zhu, and B. Zhang. Function Space Particle Optimization for Bayesian Neural Networks. In ICLR, 2019. https://arxiv.org/abs/1902.09754

Best,
Ziyu Wang

---

> ### Author Response · Authors · 2020-02-11
> **Thanks for pointing out this related work!**
>
> Hi,
>
> Thank you for leaving a comment on this! We will add a reference in the related work section for completeness!
>
> Best,
> Yeming

---

### Public Comment · ~Ke_Alexander_Wang1 · 2020-04-01
**Similar to Cheung et al., 2019**

Hi,

This's method is very similar to Cheung et al. 2019's paper: https://papers.nips.cc/paper/9269-superposition-of-many-models-into-one.pdf which also proposes sharing weights among the ensemble in a similar way and demonstrated learning up to 50 different sequential learning tasks. It might be good to add it as a reference.

---

### Decision · Program_Chairs · 2019-12-19

**Decision:**

Accept (Poster)

**Comment:**

This paper proposed an improved ensemble method called BatchEnsemble, where the weight matrix is decomposed as the element-wise product of a shared weigth metrix and a rank-one matrix for each member.  The effectiveness of the proposed methods has been verified by experiments on a list of various tasks including image classification, machine translation, lifelong learning and uncertainty modeling.  The idea is simple and easy to follow.  Although some reviewers thought it lacks of in-deep analysis, I would like to see it being accepted so the community can benefit from it.